# Prediction Accuracy of Soil Chemical Parameters by Field- and Laboratory-Obtained vis-NIR Spectra after External Parameter Orthogonalization

**DOI:** 10.3390/s24113556

**Published:** 2024-05-31

**Authors:** Konrad Metzger, Frank Liebisch, Juan M. Herrera, Thomas Guillaume, Luca Bragazza

**Affiliations:** 1Field-Crop Systems and Plant Nutrition, Agroscope, Route de Duillier 60, 1260 Nyon, Switzerland; konrad.metzger@agroscope.admin.ch (K.M.); thomas.guillaume@agroscope.admin.ch (T.G.); 2Water Protection and Substance Flows, Agroscope, Reckenholzstrasse 191, 8046 Zurich, Switzerland; frank.liebisch@agroscope.admin.ch; 3Cultivation Techniques and Varieties in Arable Farming, Agroscope, Route de Duillier 60, 1260 Nyon, Switzerland; juan.herrera@agroscope.admin.ch

**Keywords:** proximal sensing, soil moisture, soil organic carbon, clay content, total nitrogen, in situ soil spectroscopy, spectrometers, EPO, rdCV, PLSR

## Abstract

One challenge in predicting soil parameters using in situ visible and near infrared spectroscopy is the distortion of the spectra due to soil moisture. External parameter orthogonalization (EPO) is a mathematical method to remove unwanted variability from spectra. We created two different EPO correction matrices based on the difference between spectra collected in situ and, respectively, spectra collected from the same soil samples after drying and sieving and after drying, sieving and finely grinding. Spectra from 134 soil samples recorded with two different spectrometers were split into calibration and validation sets and the two EPO corrections were applied. Clay, organic carbon and total nitrogen content were predicted by partial least squares regression for uncorrected and EPO-corrected spectra using models based on the same type of spectra (“*within domain*”) as well as using laboratory-based models to predict in situ collected spectra (“*cross-domain*”). Our results show that the within-domain prediction of clay is improved with EPO corrections only for the research grade spectrometer, with no improvement for the other parameters. For the cross-domain predictions, there was a positive effect from both EPO corrections on all parameters. Overall, we also found that in situ collected spectra provided an equally successful prediction as laboratory-based spectra.

## 1. Introduction

Visible and near-infrared (vis-NIR) spectroscopy is gaining more and more attention as a fast, affordable and reliable method to measure soil fertility-related parameters [1]. The main principle of vis-NIR spectroscopy is based on the interaction of certain chemical bonds within the soil constituents with the electromagnetic radiation in the visible and near-infrared range (i.e., 350–2500 nm). As soils consist of a mixture of these spectrally active components (mainly C-H, C-O, C-N, C=O, O-H covalent bonds in clays and organic matter as well as metal-OH in clays [2]) including their overtone and combination vibrations, they result in a complex spectrum that is analyzed as a whole by means of chemometric methods. The most used multivariate statistical model is partial least square regression (PLSR), whereas other models include principal component regression (PCR), random forest (RF), regression trees (e.g., Cubist), support vector machines (SVM) and, especially for large spectral libraries, artificial neural network methods (see [3,4,5] and references therein). Over the recent decades, a considerable number of articles have been published on vis-NIR soil spectroscopy, primarily to determine the soil parameters that can be reliably predicted as well as to develop mathematical/chemometric models for the best predictions [3,5,6,7].

Considering that more portable vis-NIR spectrometers which are also economically affordable are becoming available on the market, there is an increasing interest in applying vis-NIR soil spectroscopy directly in the field. In these cases, vis-NIR soil spectra are measured in situ to directly predict soil parameters using a specific prediction model that has been developed beforehand (e.g., [8,9,10,11,12]). Such in situ spectroscopy holds a large potential for a fast and affordable soil analysis with the possibility of scanning soils at a high spatial and temporal resolution. However, this potential is also associated with new challenges that must be considered or clarified. One of the most prominent challenges is certainly the distortion in the spectra due to the presence of water (soil moisture) in the samples causing absorption peaks in the spectrum, particularly around 1415–1455 nm, 1915 nm and 2200 nm, whose intensity depends on moisture amount [2,13,14]. In addition, other distortions are caused by different soil aggregates, coarse particles, dead plant material and roots that can interfere with vis-NIR radiation [15,16,17]. Because of such distortions, generally, the prediction of soil parameters from laboratory-obtained vis-NIR spectra is better than from field-obtained (in situ) spectra because soil samples are preliminarily dried, sieved and often also ground, eliminating many issues associated with moisture content or structural heterogeneity [18,19,20]. However, the possibility of predicting soil parameters directly and reliably from field-obtained spectra has certainly the major advantage of avoiding or limiting the time spent for sample collection, transport, drying, sieving and grinding as is typical for laboratory-based spectra.

Currently, large soil spectral libraries (SSLs) are available, such as the LUCAS database for Europe, the Brazilian SSL, the Kellogg soil spectra library for the United States, the African SSL and the Global SSL, among others [21]. Most if not all these SSLs are composed of vis-NIR or mid-infrared (MIR, 2500–25,000 nm) spectra that have been obtained from dried and sieved soil samples, i.e., from laboratory-based spectroscopy [22,23]. If we consider the available SSL and the associated prediction models, it can be of practical interest to assess the possibility of predicting in situ spectra by models that are based on laboratory spectra [1,13], an option particularly helpful when prediction models created with only field-obtained spectra are not yet available. One of the major problems associated with this possibility, beside the harmonization of already existing SSLs [24], is the distortion in the spectra caused by the in situ characteristics. To deal with this problem, mathematical techniques have been developed to remove, in particular, the effect of soil moisture from the spectra so as to identify the unwanted variation and to remove such variation through transfer functions. A recent in-depth review of these techniques is given by Knadel et al. [13], in which they identified the external parameter orthogonalization (EPO) algorithm as the most used in over 50% of the investigated studies compared to other mathematical techniques such as direct standardization, orthogonal signal correction and wavelet transformation. The EPO algorithm was initially developed by Roger et al. [25] to remove the influence of temperature on the NIR measurements of sugar content in fruits and by Minasny et al. [15] to remove the influence of soil moisture on soil organic carbon (SOC) prediction.

The EPO algorithm is an unmixing model dividing the spectral dataset (**X**) into a part containing the useful information (**XP**), a part containing the unwanted variation (**XQ**) and a residual part (**R**) [15,26] as follows:**X** = **XP** + **XQ** + **R**(1)

In this matrix form, the spectral dataset **P** and **Q** are projection matrices of the useful and the unwanted parts of the spectra, respectively. The goal is to obtain the projection matrix **P** to estimate the useful part **X^*^** of the spectra according to Equation (2), with **I** being the identity matrix:**X*** = **XP** = **X**(**I** − **Q**)(2)

To estimate the projection matrix **Q** for the uninformative part of the spectra (orthogonal to the desired spectra), the first principal component of the difference matrix **D**, i.e., the difference between the spectra with the unwanted variation and the laboratory grade spectra, can be used.

Once the projection matrix **P** is calculated, both the laboratory-based SSL and the in situ spectra are multiplied by **P**, thus making the spectral libraries comparable. This procedure should potentially allow for the prediction of an EPO-corrected in situ spectrum with calibration models based on a EPO-corrected laboratory SSL [13].

The basis on which the difference matrix **D** for the EPO algorithm is calculated varies between studies. In the majority of the studies, the spectra of remoistened samples, after having been sieved and dried, are used with the intention of controlling the moisture levels and to feed different moisture levels of the same soil into the EPO (see Knadel et al. [13] and references within). Other studies used field-obtained spectra and their corresponding dried and sieved laboratory spectra to calculate **D**, considering that the exact moisture content of the sample is not necessary to successfully calculate the EPO projection matrix [27], thus reducing the number of processing steps. In a third methodology, as proposed by Chakraborty et al. [28], the authors used both field moist and rewetted spectra to compare the EPO transformed results to those obtained from an EPO transformation matrix based solely on rewetted spectra. In this case, they found that the dataset corrected with spectra of field moist samples had a higher RPIQ (ratio of performance to interquartile range) compared to the correction with rewetted samples.

When the EPO algorithm is applied to field-obtained spectra in order to directly predict field-obtained spectra, it has been shown that this transformation reduced the RMSEP (root mean squared error of prediction) for the clay content compared to untransformed spectra, suggesting a potential improvement of in situ predictions [29], but no information is available for other soil parameters. To fill this knowledge gap, we want to assess the potential of the EPO algorithm to improve the predictions of soil organic carbon (SOC), total nitrogen (Ntot) and clay content, based on a set of vis-NIR spectra measured both in the field and in the laboratory. More specifically, we want to corroborate the following hypotheses:Without any EPO correction, laboratory-obtained spectra provide better prediction models if compared to the field-obtained spectra (hereafter called *within-domain* prediction).The prediction of soil parameters using field-obtained spectra with models based only on field-obtained spectra (=in situ *within-domain* prediction) results in better prediction accuracy after EPO correction.Predicting in situ spectra from models that have been created using laboratory-obtained spectra results in a lower model performance, but with the EPO correction, the model performance can be improved (hereafter called *cross-domain* prediction).From a methodological point of view, the *cross-domain* prediction accuracy does not differ if we use finely ground soil samples as a baseline for the EPO transfer matrix or if we use soil samples that have been simply dried and sieved (<2 mm).The *within-domain* prediction for field-obtained spectra is always better than a cross-domain prediction.

## 2. Materials and Methods

### 2.1. Field Sampling

The agricultural fields used for this study were sampled in 2021 and are described in detail in Metzger et al. [12]. Briefly, nine experimental fields were selected, with a variety of treatments regarding tillage intensity and depth (no-till to conventional ploughing), fertilization type (ranges of mineral and organic fertilizers) and added biomass (cover crops, fragmented ramial woodchips) in Calcaric Cambisol (7 fields), Cambisol (1 field) and Luvisol (1 field). In each experimental plot, one sample was taken with a 20 cm helical Edelman Auger (Eijkelkamp, Giesbeek, The Netherlands) resulting in 134 sampled points (soil samples) with corresponding spectra and laboratory analyses.

### 2.2. Spectrometers

For the spectral analysis, two different types of portable spectrometers were used (Table 1). As an example of a research-grade spectrometer, the PSR+3500 (PSR, Spectral Evolution, Haverhill, MA, USA) with a spectral range of 350–2500 nm and a resampled output resolution of 1 nm was used. As an example of a consumer-grade, relatively new and portable near-infrared scanner, the NeoSpectra Scanner (NEO, SiWare, Cairo, Egypt) with a micro-electro-mechanical-systems (MEMS) Fourier-transform Michaelson interferometer was used. The NEO spectrometer collects spectra in the range of 1350–2500 nm and reports with a resampled resolution of ca. 13.5 wavenumbers (ca. 2.5–8.8 nm).

### 2.3. Laboratory Analyses

For the chemical laboratory analysis, the sample was taken out of the Edelmann auger after spectra acquisition in the field and transported into the laboratory where it was dried at 40 °C for 24 h and sieved to <2 mm. The samples were then analyzed for organic carbon (SOC) content by sulfochromic wet oxidation (Walkley–Black method, NF ISO 14235 [30]), total nitrogen (Ntot) (Elemental Analyzer, NF ISO 13878 [31]) and clay content (sieve–pipette method) [32]. The water content of the fresh samples was determined gravimetrically by taking a subsample and weighing it before and after drying at 105 °C for 24 h.

### 2.4. Soil Spectra Acquisition in the Field

After extraction of the soil sample in the field with the helical Edelmann auger, the soil surface of one side of the auger was cut with a knife to obtain a flat surface, taking care not to smear the soil. Soil spectra were recorded with each of the spectrometers on five points covering the entire 20 cm length of the core. With the PSR spectrometer, a contact probe with an active light source (5 W tungsten halogen lamp) and 10 mm spot size was used. For the NEO spectrometer, the entire instrument was placed directly onto the soil sample with the measuring widow on the measuring point and the housing of the instrument covering the incoming light (Figure 1a,b). For the PSR spectrometer, a Zenith Polymer^®^ reflectance panel (SphereOptics, Herrsching, Germany) was used to collect the white reference before each sampling point, whereas for the NEO spectrometer the inbuilt reflectance panel was used before scanning each sampling point.

### 2.5. Laboratory Scanning of Sieved and Ground Soil Samples

The dried and sieved soil samples (hereafter also called “sieved-only samples”) were placed in a Petri dish (diameter 100 mm). The surface was flattened with the lid of the Petri dish before the soil was scanned in three replicate scans with the contact probe of the PSR spectrometer while rotating the sample 120° after each scan. The white reference was recorded every 20 scans. For the NEO spectrometer, the soil was placed in a weighing dish and the surface was smoothed with a blade. The weighing dish was then placed into a cardboard frame on which the spectrometer was placed, thus ensuring good contact between the measuring spot and the soil, while excluding the ambient light. The sample was scanned in triplicate and rotated 90° after each scan. The white reference was scanned every 20 scans.

For the finely ground soil samples, a subsample was milled with a ball mill (Retsch, Haan, Germany) and filled into the 100 mm diameter Petri dish. As the finely ground soil sticks to the rubber ring of the fore optic of the PSR contact probe, the soil sample was scanned with a bare fiber with a 25° field-of-view following a protocol for measuring soils [33,34]. In brief, the bare fiber was placed 8 cm above the surface of the soil sample which was illuminated at a 30° zenith angle with two 100 W tungsten-halogen lamps in an otherwise dark room. The sample was again rotated 120° for the three replicate scans and the white reference was scanned after each batch of five samples. For the NEO spectrometer, the same setup as for the sieved samples was used, as there was no disturbance due to the flat scanner surface as opposed to the rubber ring of the PSR contact probe (Figure 1c,d).

### 2.6. Spectra Processing

Initially, the spectral replicates of each soil sample were checked for outliers using the spectral standard deviation, i.e., the standard deviation for each wavelength across the replicates, then the standard deviation thereof across all wavelengths). Accordingly, if the respective value of standard deviation was >0.01 the replicates were checked and spectra that were considered an outlier were removed [35]. After the removal of potential outliers, the average reflectance value was then calculated from the residual replicate scans. On average, for the set of 5 replicates for the field-based scans there were between zero to 1 outlier for the each of the two instruments, whereas for the set of 3 replicates for the laboratory-based scans there were always zero outliers. In the case of the PSR spectrometer, spectra for all 134 sampling points were obtained, whereas for the NEO spectrometer the spectra from 40 points had to be removed due to a sensor malfunction, resulting in a final dataset for the NEO spectrometer of 94 spectra (soil samples). The spectra from the NEO spectrometer were transformed from wavenumbers [cm^−1^] into wavelengths [nm] and resampled to 2 nm with the “prospectr” R package [36]. All spectra were then transformed from reflectance (R) into absorbance (A) with A = log (1/R) followed by a Savitzky–Golay smoothing [37] and a standard normal variate (SNV) preprocessing step [38]. The so created datasets (*n* = 134 for the PSR and *n* = 94 for the NEO spectrometer) were then split into a calibration set (PSR: *n* = 95, NEO: *n* = 71) and a validation set (PSR: *n* = 39, NEO: *n* = 23) with the Kennard–Stone algorithm also with the “prospectr” package [36,39]. A visualization of the first two principal components of the PSR spectra and the selection of calibration and validation set is given in Figure 2 as example. All data treatment was executed using the R software version 4.3.2 [40] according to Wadoux et al. [41].

### 2.7. EPO Correction Procedure

The EPO correction matrix was calculated based on 10 samples, which were selected to cover the maximum range of SOC and clay content (see Table 2). The difference matrix **D** was then calculated in two different ways and used to form the EPO correction matrices, i.e., for the ***EPO_sieved*** the difference between the spectra of the field moist samples and the sieved-only samples was calculated, whereas for the ***EPO_fine*** the difference between the spectra of the field moist samples and the finely ground samples was calculated. For both the EPO matrices, the optimum number of factors was determined by using the maximum Wilk’s Λ [26].

The ***EPO_sieved*** matrix (=EPO_s) was then used to correct both the in situ collected spectra (***field moist***) subset and the laboratory-based, sieved-only subset (***lab_sieved***) on the other hand. Equally, the ***EPO_fine*** matrix (also = EPO_f) was used to correct the field-obtained (***field moist***) spectra and the laboratory-based, sieved and finely ground spectra (***lab_fine***). This resulted in four new EPO-corrected datasets, each consisting of a calibration (***_cal***) and a validation (_***val***) subset, i.e., ***sieved_EPO_s*** and ***field_EPO_s*** (=respectively, the ***lab_sieved*** spectra and the ***field moist*** spectra corrected with ***EPO_sieved*** matrix) and ***fine_EPO_f*** and ***field_EPO_f*** (=respectively, the ***lab_fine*** spectra and ***field moist*** spectra corrected with the ***EPO_fine*** matrix). An overview over the EPO-corrected datasets and the applied models is given in Figure 3.

### 2.8. Model Calibration

Seven calibration subsets were available for modelling from each of the two spectrometers: three uncorrected (raw) datasets (***field moist***, ***lab_sieved*** and ***lab_fine***) and four EPO-corrected datasets (***sieved_EPO_s***, ***field_EPO_s***, ***fine_EPO_f*** and ***field_EPO_f***). Each of the subsets was then used to calibrate a model for clay, SOC and Ntot using partial least squares regression (PLSR), a method capable of working with relatively small datasets (see Esbensen and Swabrick ref. [42], in a 100-times repeated double cross-validation (rdCV) using the “chemometrics” R package [43,44,45]. Briefly, with the rdCV, the model performance is evaluated in two loops. In the inner loop, the optimum number of latent variables (LVs) is determined based on the minimum standard error of prediction (SEP) of a 10-fold cross-validation, which is in turn used to parametrize the outer loop in a 4-fold cross validation. This nested loop is then repeated 100 times with random splits of the cross-validation loops and, finally, the optimum number of LVs and the model performance indicators such a R^2^ (coefficient of determination) and RMSE (root mean squared error) are calculated as the average of each of the 100 repetitions.

### 2.9. Model Validation

The models were then validated by predicting the selected soil parameters using the spectra from the validation set in a PLSR with the optimum number of LVs as determined for each parameter and calibration set in the rdCV. For the *within-domain* prediction, the models created on one calibration set were applied to its corresponding validation set. This resulted in models from raw spectra for the ***lab_sieved*** samples, the ***lab_fine*** samples and the ***field moist*** samples, as well as for the EPO corrected ***sieved-EPO_s***, ***field-EPO_s***, ***fine-EPO_f*** and ***field_EPO_f*** spectra. For the *cross-domain* prediction, the models created from lab-obtained spectra were applied to the field-obtained validation sets, i.e., for the raw spectra ***sieved_cal-field moist_val*** and ***fine_cal-field moist_val***; for the EPO-corrected spectra ***sieved_EPO_s_cal-field_EPO_s_val*** and ***fine_EPO_f_cal-field_EPO_f_val*** (see Figure 3).

The predictive performance of the validation was then evaluated based on the R^2^, the root mean squared error of prediction (RMSEP), the ratio of performance to inter-quartile distance (RPIQ = IQR/RMSEP, IQR = inter-quartile range of the laboratory measured soil parameter [46]), the bias and Lin’s concordance correlation coefficient (CCC) [47].

## 3. Results

### 3.1. Laboratory Analyses of Soil Samples

Based on laboratory analyses, the water content of all soil samples ranged from 8.5% to 37.3%, the SOC from 0.74% to 3.67%, the Ntot from 0.07% to 0.4% and the clay content from 11.4% to 55.5%. Similarly, the values of the parameters for the set of soil samples used for the calibration–validation and for the EPO correction span a range of variability comparable to the entire data set (Table 2).

### 3.2. Spectral Correction with EPO

The best correction (i.e., highest Wilk’s Λ, least separation between field-obtained and laboratory-obtained spectra) was reached with five and seven factors for, respectively, the sieved and the finely ground soil spectra in the case of the PSR spectrometer. On the other hand, for the NEO spectrometer, the maximum Λ was reached with five factors for both the sieved and the finely ground soil spectra.

A visual comparison for an exemplary soil sample between the uncorrected and the EPO-corrected spectrum is given in Figure 4 for both the PSR and the NEO spectrometers. In the uncorrected spectra, we can clearly see the differences between the dried and field moist spectra, especially around the typical water related peaks at 1400 and 1900 nm in combination with absorption of other organic matter groups (amines, carboxylic groups around 1900 nm) or kaolinite (1400 nm) as well as a small water-related peak around 2200 nm [2,48,49] (Figure 4a,c). In the EPO-corrected spectra (Figure 4b,d), however, the differences are drastically reduced. For the PSR spectrometer (top), there are still few distinct peaks, again at around 1400 and 1900, among others, with the absorbance ranging from <−1 to >1. For the NEO spectrometer, on the other hand, there are many other small peaks and the absorbance ranges from <−0.2 to >0.2

### 3.3. Within-Domain Modelling Results

The prediction results for the uncorrected and the EPO-corrected spectra with the different combinations of datasets and domains are given in Table 3 for the PSR and the NEO spectrometers.

Based on the RPIQ, for the uncorrected spectra the best validation results for the PSR and the NEO spectrometers were obtained using spectra from laboratory soil samples for the clay content, more specifically from ground samples for the PSR (RPIQ = 3.47) and sieved for the NEO (RPIQ = 3.11). For Ntot, the best validation results were obtained for field-obtained spectra with RPIQ values of 2.21 and 2.44 for the PSR and the NEO spectrometer, respectively. Similarly, also for the SOC the best validation results were observed from field-obtained spectra with RPIQ values of 2.23 and 1.56 for the PSR and the NEO spectrometer, respectively (Table 3). A plot of laboratory and predicted values from field-obtained, sieved and ground spectra for clay, Ntot and SOC for each spectrometer is given in Figure 5.

The comparison between *within-domain* model results from uncorrected and EPO-corrected field-obtained spectra with the PSR spectrometer shows that there is an improvement of the validation results for clay for both EPO-correction matrices (EPO_s, EPO_f). In the case of both Ntot and SOC, the improvement of the validation results can be obtained only if the EPO-correction matrix based on finely ground samples (EPO_f) is applied (Table 3, Figure 6). When soil spectra are collected with the NEO spectrometer, the EPO-correction does not show any improvement of the validation results for all the three soil parameters.

### 3.4. Cross-Domain Modelling Results

The prediction of field-obtained spectra from models that have been created with laboratory-obtained spectra is clearly improved after EPO-correction for all the three soil parameters and for both the instruments (Table 3). Such improvement can be observed when both the EPO-correction matrices based on sieved (***EPO_sieved***) and ground (***EPO_fine***) soil samples are applied (Table 3, Figure 6 and Figure 7). Concerning the type of EPO-correction matrix, in the case of the PSR spectrometer, the correction with the matrix based on sieved-only soil samples (***EPO_sieved***) provides higher prediction accuracy (higher RPIQ) for all the three soil parameters (RPIQ 0.12, 0.19 and 0.15 for the uncorrected versus 1.14, 1.61 and 1.45 for the ***EPO_sieved***-corrected predictions of clay, Ntot and SOC, respectively). Differently, the application of the EPO-matrix based on ground soil samples (***EPO_fine***) provides better prediction accuracy for the NEO spectrometer for all the three soil parameters (RPIQ 0.19, 0.48 and 0.59 for the uncorrected versus 1.3, 2.06 and 1.26 for the with ***EPO_fine*** corrected predictions of clay, Ntot and SOC, respectively) (Table 3, Figure 6).

In Figure 6, the RPIQ values for raw and EPO-corrected field moist spectra for *within-domain* and *cross-domain* predictions are reported, whereas in Figure 7, the plot of laboratory and *cross*-domain predicted values with and without the EPO-correction for clay, Ntot and SOC for each spectrometer can be observed.

## 4. Discussion

### 4.1. Effect of Soil Sample Processing on Uncorrected, Within-Domain Spectra Predictions

When comparing the uncorrected *within-domain* predictions for the PSR spectrometer, the best results in terms of RPIQ for clay are obtained by using ground samples (RPIQ 3.47), then by sieved samples (RPIQ 3.06) and, at the last, by field moist (in situ) samples with an RPIQ of 2.08, which corresponds to only roughly 60% of the best prediction accuracy. For Ntot and SOC, on the other hand, there are not strong differences among the RPIQ values for sieved and field-moist samples. This conclusion goes against our first hypothesis that the best model results are reached using laboratory-obtained spectra (sieved or sieved and finely ground). A similar trend is found for the NEO spectrometer. Indeed, the predictions for clay content from the laboratory-obtained spectra had the highest RPIQ for the sieved and for the sieved and finely ground soil samples, respectively, whereas the prediction from field-obtained spectra resulted in a 50% reduction in RPIQ (Table 3). For Ntot and SOC, the differences are lower. For SOC, the RPIQ was always below 2, with the result of the model based on the ground samples almost reaching the threshold of a successful model of 1.89 as proposed by Ludwig et al. [50]. For the other treatments (i.e., sieved-only and field moist), the RPIQ remained equal to about 1.5 (Figure 6).

The use of in situ spectra to create prediction models for Ntot and SOC did not reduce the model performance compared to models that are based on laboratory spectra. This is a promising result for the development of field-based calibration models to predict, at least, Ntot and SOC. This means that with a small dataset, there is the possibility that calibration models based on in situ spectra can outperform laboratory-based models, a result that was also reported by Cambou et al. [51]. In other comparative studies, however, the laboratory-based predictions outperformed the field-based models [52], so further research is necessary to better examine this phenomenon.

Our results also showed that for clay content, the moisture had a strong effect on the prediction models and that laboratory-based spectra provide better predictions for both spectrometers, an effect clearly due to the interference with water that masks the clay absorptions [13,53,54].

The comparison of the uncorrected *within-domain* predictions shows that the gains in RPIQ obtained by ground soil samples are marginal (highest RPIQ improvement ca. 12% for clay with the PSR spectrometer and ca. 16% for the NEO spectrometer). This result leaves open the question if the labor-intensive work of grinding the soils is justified for such levels of accuracy improvement.

### 4.2. Effect of EPO Correction for Field-Obtained Spectra Prediction

The improvement of the prediction accuracy after the EPO correction to field-obtained spectra, a result previously reported for clay by Veum et al. [29], is confirmed in our study mainly for the spectra collected with the PSR spectrometer. Here, the ***EPO_fine*** correction improved the uncorrected RPIQ of the clay prediction by 87% and the ***EPO_sieved*** correction by 46% (Table 3). For Ntot and SOC, there were also slight improvements of the RPIQ, but only for the ***EPO_fine*** correction, whereas the ***EPO_sieved*** correction did not improve the RPIQ of these parameters (Table 3). The RMSEP values in our study (1.6% and 0.21% for the best field moist prediction for clay and SOC, respectively) are comparable to the values that Veum et al. [29] reported in their study (7.78% and 0.26% for clay and SOC in field moist predictions, respectively).

In the case of the NEO spectrometer, we did not observe any improvement with any of the two EPO correction matrices. At the best, it led to the same results for the clay content with the ***EPO_sieved*** correction and for the Ntot with the ***EPO_fine*** correction (Table 3).

Overall, our second hypothesis that the EPO-correction improves in situ *within-domain* predictions is only true for the clay content. An explanation for this could be the fact that the clay content is strongly influenced by the water content, which makes the correction more effective [54]. In the case of the NEO spectrometer, the EPO correction did not improve the predictions, which may be caused by the shortened wavelength range (1350–2500 nm), or it may be also due to a less efficient EPO correction leaving the corrected spectra noisier compared to the spectra from the PSR spectrometer (see Figure 4).

### 4.3. Effect of the EPO Correction on Cross-Domain Prediction

When comparing the *cross-domain* predictions, i.e., predicting field-obtained spectra using models based on laboratory-obtained spectra, there is a clear effect of the EPO-correction which leads to a corrected RPIQ which is 8- to 13- fold the uncorrected RPIQ for the PSR spectrometer and 3- to 8-fold the uncorrected RPIQ for the NEO spectrometer (Table 3, Figure 7). This shows that the EPO correction is able to reduce the distorting impacts due to soil moisture and heterogeneity when training *cross-domain* models [13,15,55]. The effect of not using the EPO-correction results, indeed, in unrealistic predicted values, particularly visible for the clay content, where values >100% were predicted. However, our results also show that the *cross-domain* predictions after the EPO correction are still inferior compared to the *within-domain* predictions based on field moist spectra for all the selected soil parameters and for both the instruments (Figure 6). This is highlighted in the RPIQ of the EPO-corrected predictions, which is below the widely used RPIQ threshold (i.e., 1.89) for a successful prediction [50,56]. The only successful prediction based on that threshold would be the prediction of Ntot with the NEO spectrometer (RPIQ 2.06).

Despite the unsatisfactory RPIQ for most of the *cross-domain* predictions, other performance indicators of our models are similar to those of other studies, such as that of Ge et al. [54], where they showed an improvement of the prediction accuracy when the EPO correction is applied for clay more than for SOC prediction. In their study, they obtained RMSE values of 90 g/kg and 7.3 g/kg for clay and SOC, respectively, compared to the RMSE of our study of 57 g/kg and 3.3 g/kg for the PSR spectrometer and 56 g/kg and 3.1 g/kg for the NEO spectrometer, for clay and SOC, respectively. Much better results in terms of RMSE were reached by Minasny et al. [15] who obtained values of up to 0.2 g/kg and by Mirzaei et al. [57] in terms of RPIQ, who improved the RPIQ to 2.93 for clay and 3.44 after EPO correction.

One reason for the low RPIQ of our models may be the low number of samples used for both the model calibration (*n* = 95 for the PSR spectrometer and 71 for the NEO spectrometer) and especially for the creation of the EPO matrix (*n* = 10). Minasny et al. [15], for example, recommended using at least 60 samples to create the EPO correction matrix. Indeed, in their study, the range of SOC and clay values was wider in the EPO set as well as in the calibration and validation set compared to the samples in our study.

### 4.4. Practical Applications of Within-Domain and Cross-Domain Predictions

When the two methods to create the EPO correction matrix (i.e., the ***EPO_sieved***, based on sieved-only samples and the ***EPO_fine***, based on sieved and ground samples) are compared, the ***EPO_sieved*** correction led to equal or higher RPIQ values for all the selected chemical parameters in the case of the PSR spectrometer. However, the ***EPO_fine*** correction resulted in higher RPIQ values in the case of the NEO spectrometer. As the RPIQ values of the EPO-corrected models are low overall, it is challenging to give clear advice to the potential end user as to which EPO-method correction to use. However, if the differences between the EPO corrections remain as low as in our present study, it is questionable if it is worth the extra effort of finely grinding the soil samples to calculate the **D** matrix for the EPO calculation, thus confirming our fourth hypothesis.

The same applies for the uncorrected laboratory-based predictions where there are slightly better model performances for some parameters (i.e., clay, Ntot and SOC with the PSR spectrometer and SOC with the NEO spectrometer). The additional effort of finely grinding the samples is very likely not worth it, especially when vis-NIR spectroscopy should be a fast alternative to predict soil parameters. The loss in accuracy might get counterbalanced by a higher number of samples, a topic which is in urgent need of deeper analysis.

In our study based on a small dataset, it is obvious that the *within-domain* models result in higher prediction accuracy when compared to *cross-domain* models, confirming our fifth hypothesis. Based on these results, one could argue if it makes more sense to use only in situ field moist spectra when a local database is built from scratch. In cases where only an SSL based on laboratory-spectra is available, the accuracy in predicting field moist spectra definitely improves when the in situ spectra are corrected with the EPO.

## 5. Conclusions

In this study, we were able to show that, in the case of *within-domain* predictions, there is hardly any difference between using dried, sieved-only soil samples and dried, sieved and finely ground soil samples. In addition, for SOC and Ntot the prediction models based on field-obtained (= in situ) spectra show the same level of RPIQ of the laboratory-obtained spectra. Instead, the prediction of the clay content showed a deterioration of RPIQ when in situ based models are used. However, in the case of *within-domain* prediction of the clay content based on field-obtained spectra, the EPO correction (especially the ***EPO_fine***) improved considerably the model prediction accuracy with a very high RPIQ particularly for the PSR spectrometer. On the other hand, for SOC and Ntot the EPO correction did not improve the prediction accuracy for both the spectrometers.

When it is necessary to set up a *cross-domain* prediction, the EPO correction can definitively improve the model performance allowing to predict field-obtained spectra using models that are based on laboratory-obtained spectra. Overall, we saw little effect in using ground samples as baseline for the EPO transfer matrix compared to sieved-only samples. The EPO correction for *cross-domain* prediction was effective for both the spectrometers.

The overall results of this study show that the use of in situ vis-NIR soil spectroscopy has great potential and while for Ntot and SOC there is no benefit of using EPO correction, the in situ prediction of clay content can be improved with the EPO correction, particularly when a spectrometer with high spectral resolution is used. Our results confirm that vis-NIR spectroscopy can be a reliable methodology for obtaining soil fertility-related information directly in the field.

## Figures and Tables

**Figure 1 sensors-24-03556-f001:**
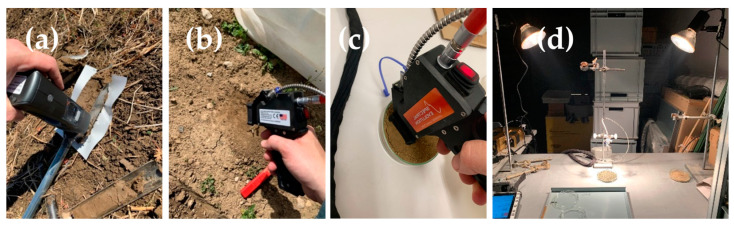
Scanning setup for in situ spectra acquisition with the NeoSpectra Scanner (**a**) and with the PSR+3500 contact probe (**b**). The setup for laboratory spectra acquisition of only sieved (lab_sieved) soil samples (**c**) and finely ground (lab_fine) soil sample (**d**) with the PSR+3500 bare fiber.

**Figure 2 sensors-24-03556-f002:**
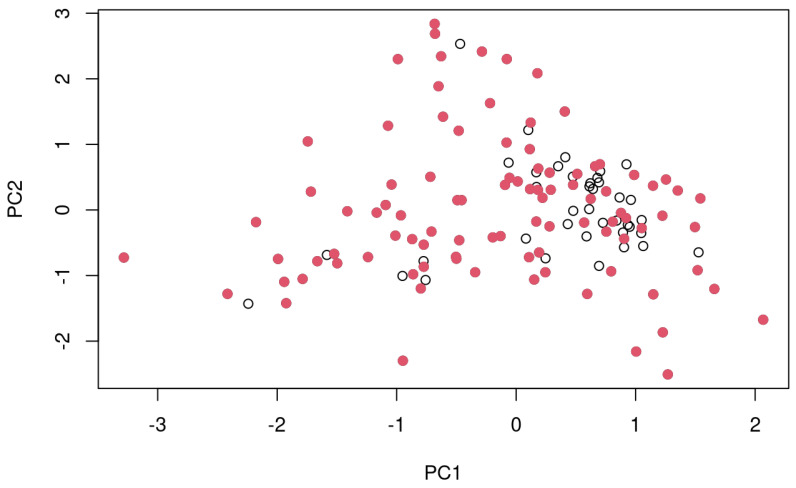
Visualization of the first two principal components (PC1 and PC2) of the dataset with the calibration (red dot) and the validation (circle) samples as selected by the Kennard–Stone algorithm for the PSR spectrometer. The ratio calibration/validation was of 70/30.

**Figure 3 sensors-24-03556-f003:**
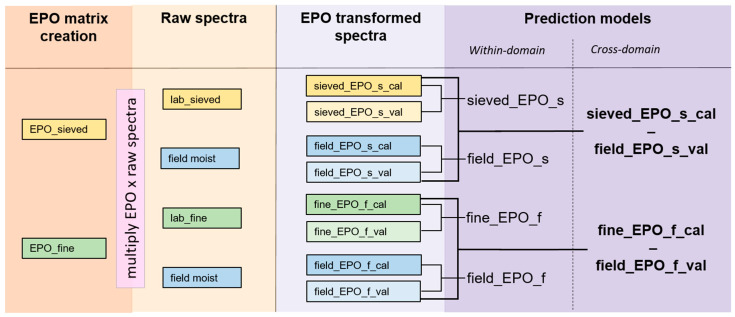
Overview of the EPO correction and the models created with the EPO-corrected spectra. This scheme applies to the spectra collected with both the spectrometers. See the main text for abbreviations.

**Figure 4 sensors-24-03556-f004:**
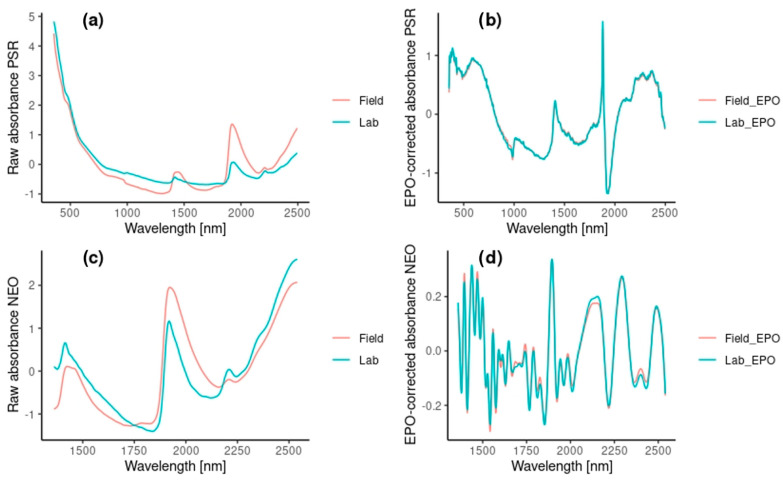
Raw absorbance spectra (**a**,**c**) and EPO-corrected absorbance spectra (**b**,**d**) from an exemplary soil sample for the PSR spectrometer (**a**,**b**) and the NEO spectrometer (**c**,**d**) for the sieved-only (lab: green) and in situ (field: red) scans. In (**a**,**c**), absorption peaks of H_2_O are clearly visible around 1400 nm, 1900 nm and 2250 nm. Note the difference scale of the *y*-axis.

**Figure 5 sensors-24-03556-f005:**
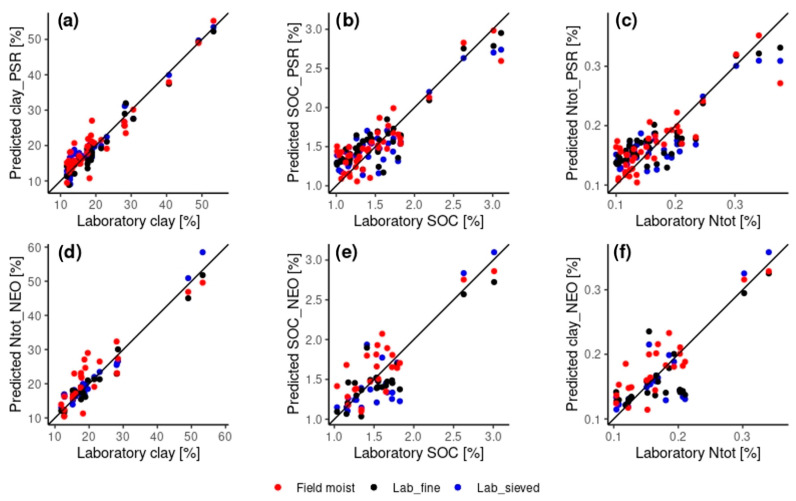
Laboratory versus predicted values from the validation set for the within-domain predictions for clay (**a**,**d**), SOC (**b**,**e**) and Ntot (**c**,**f**) for the PSR (**a**–**c**) and the NEO (**d**–**f**) spectrometer including the 1:1 line. The predicted values for the field-obtained (=field moist) model are displayed in red, for the lab_fine in black and for the lab_sieved in blue. For an overview of the modeling process, see Figure 3. For the level of accuracy of prediction, see Table 3 for the modelling “field moist”, “lab_sieved” and “lab_fine” for the two spectrometers and the three parameters.

**Figure 6 sensors-24-03556-f006:**
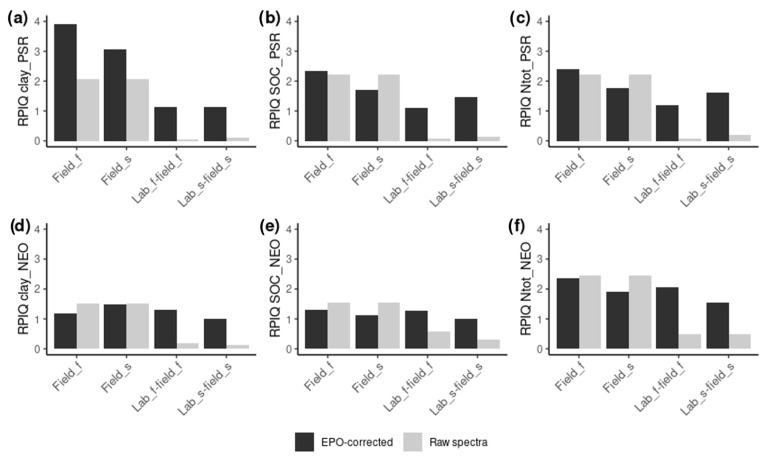
RPIQ values for uncorrected (raw) spectra (grey) and correspondent EPO-corrected spectra (black) for within domain predictions (field_f and field_s) and cross-domain predictions (lab_f-field_f and lab_s-field_s) for the PSR (**a**–**c**) and the NEO (**d**–**f**) spectrometers and for clay (**a**,**d**), SOC (**b**,**e**) and Ntot (**c**,**f**). The values of the raw field-obtained spectra (field_s and field_f) are the same and repeated for a better comparison.

**Figure 7 sensors-24-03556-f007:**
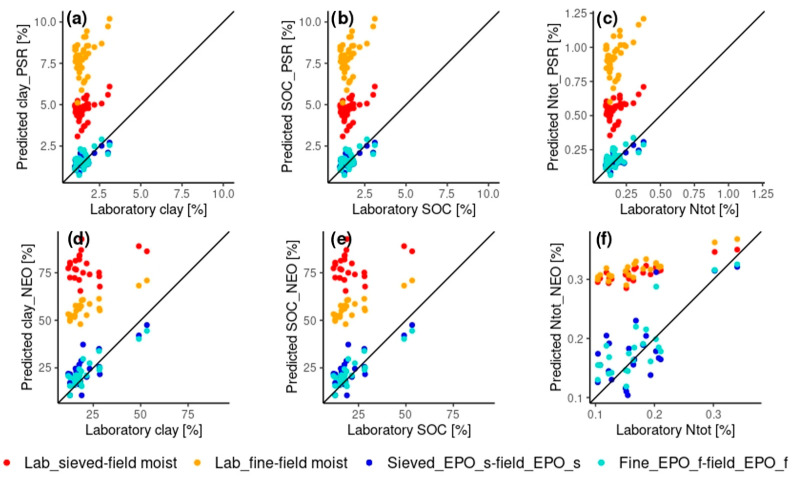
Laboratory versus predicted values from the validation set for the cross-domain predictions for clay (**a**,**d**), SOC (**b**,**e**) and Ntot (**c**,**f**) for the PSR (**a**–**c**) and the NEO (**d**–**f**) spectrometers including the 1:1 line. The predicted values for the uncorrected predictions are displayed in red for the *lab_sieved-field moist* model and in orange for the *lab_fine-field moist* model, whereas for the EPO-corrected models the predicted values are in dark blue for the *sieved_EPO_s-field_EPO_*s model and in light blue for the *fine_EPO_f-field_EPO_f* model. For an overview over the modeling process, see Figure 3 and Table 3.

**Table 1 sensors-24-03556-t001:** Specifications of the used spectrometers.

Instrument	PSR+3500 (Spectral Evolution, Haverhill, MA, USA)	Neospectra Scanner (Siware Systems, Cairo, Egypt)
Spectral range	350–2500 nm	1350–2500 nm
Spectral resolution	2.8–8 nm	16 nm
Spot size (contact probe)	10 mm	10 mm
Field of view (bare fiber)	25°	-
Detector type	Si photodiode array (350–1000 nm) InGaAs photodiode array (970–1910 nm, 1900–2500 nm)	FT-NIR optical MEMS Michaelson interferometer and InGaAs photodetector

**Table 2 sensors-24-03556-t002:** Summary statistics of the laboratory analyses for the entire dataset (all), the EPO training set (EPO), the calibration set and the validation set for water content (wc), soil organic carbon (SOC), soil total nitrogen (Ntot) and clay content.

	**All Soil Samples, *n* = 134**				**EPO Soil Sample Set, *n* = 10**			
	**wc [%]**	**SOC [%]**	**Ntot [%]**	**Clay [%]**	**wc [%]**	**SOC [%]**	**Ntot [%]**	**Clay [%]**
**MIN**	8.54	0.74	0.07	11.35	16.08	0.74	0.07	11.35
**Q1**	18.00	1.26	0.12	14.03	19.19	1.39	0.14	17.96
**MEDIAN**	21.03	1.54	0.17	19.33	23.23	1.71	0.21	20.78
**MEAN**	22.78	1.74	0.19	24.51	23.79	1.82	0.20	24.22
**Q3**	27.83	2.20	0.24	29.14	27.07	2.14	0.23	28.39
**MAX**	37.31	3.67	0.40	55.46	36.73	2.95	0.35	55.46
	**Calibration Set, *n* = 95**				**Validation Set, *n* = 39**			
	**wc [%]**	**SOC [%]**	**Ntot [%]**	**Clay [%]**	**wc [%]**	**SOC [%]**	**Ntot [%]**	**Clay [%]**
**MIN**	10.27	0.74	0.07	11.35	8.54	1.01	0.10	11.82
**Q1**	18.98	1.30	0.13	16.89	17.50	1.17	0.12	12.74
**MEDIAN**	22.08	1.57	0.18	25.19	18.37	1.40	0.14	16.04
**MEAN**	23.71	1.84	0.20	26.65	20.52	1.51	0.16	19.30
**Q3**	28.72	2.53	0.29	31.58	23.45	1.66	0.19	19.26
**MAX**	37.31	3.67	0.40	55.46	33.47	3.11	0.38	53.28

**Table 3 sensors-24-03556-t003:** Prediction results for uncorrected and EPO-corrected spectra for the PSR and the NEO spectrometers for clay, Ntot and SOC. The cross-domain predictions are in italics. The naming of the EPO-corrected modelling is in accordance with Figure 3.

			Spectral Evolution PSR+3500 (PSR)					SiWare NeoSpectra Scanner (NEO)				
EPO Corrected	Modelling	Parameter	RMSEP	R^2^	Lin’s CCC	RPIQ	Bias	RMSEP	R^2^	Lin’s CCC	RPIQ	Bias
**YES**	sieved_EPO_s	clay [%]	2.47	0.93	0.97	2.64	−0.46	2.81	0.93	0.97	2.43	−0.27
**NO**	lab_sieved	clay [%]	2.14	0.95	0.98	3.06	−0.54	2.2	0.95	0.98	3.11	−0.71
**YES**	fine_EPO_f	clay [%]	2.8	0.92	0.96	2.33	−0.66	3.5	0.89	0.93	1.95	0.50
**NO**	lab_fine	clay [%]	1.88	0.96	0.98	3.47	0.30	2.29	0.95	0.97	2.98	0.77
**YES**	field_EPO_s	clay [%]	2.14	0.95	0.97	3.05	−0.62	4.56	0.81	0.9	1.5	−2.06
**NO**	field moist ^(1)^	clay [%]	3.14	0.89	0.95	2.08	−0.94	4.51	0.81	0.9	1.52	−1.67
**YES**	field_EPO_f	clay [%]	1.67	0.97	0.98	3.9	−0.38	5.82	0.68	0.83	1.17	−2.29
**NO**	field moist ^(1)^	clay [%]	3.14	0.89	0.95	2.08	−0.94	4.51	0.81	0.9	1.52	−1.67
** *YES* **	sieved_EPO_s-field_EPO_s	clay [%]	5.71	0.65	0.83	1.14	−1.63	6.94	0.55	0.75	0.99	−2.39
** *NO* **	lab_sieved-field moist	clay [%]	56.36	−33.09	0.05	0.12	−55.69	58.11	−30.54	0.01	0.12	−57.12
** *YES* **	fine_EPO_f-field_EPO_f	clay [%]	5.73	0.65	0.84	1.14	−2.20	5.26	0.74	0.84	1.3	−1.61
** *NO* **	lab_fine-field moist	clay [%]	118.09	−148.62	0.02	0.06	−116.7	35.94	−11.07	0.07	0.19	−35.31
**YES**	sieved_EPO_s	Ntot [%]	0.03	0.75	0.85	2.22	−0.01	0.03	0.8	0.89	2.85	0.00
**NO**	lab_sieved	Ntot [%]	0.03	0.71	0.81	2.07	0.00	0.03	0.64	0.83	2.11	0.01
**YES**	fine_EPO_f	Ntot [%]	0.03	0.74	0.84	2.18	−0.01	0.03	0.76	0.85	2.57	0.00
**NO**	lab_fine	Ntot [%]	0.03	0.74	0.84	2.18	−0.01	0.04	0.62	0.79	2.06	0.01
**YES**	field_EPO_s	Ntot [%]	0.04	0.6	0.78	1.76	−0.01	0.04	0.56	0.8	1.92	−0.01
**NO**	field moist ^(1)^	Ntot [%]	0.03	0.74	0.85	2.21	−0.01	0.03	0.73	0.86	2.44	−0.01
**YES**	field_EPO_f	Ntot [%]	0.03	0.78	0.86	2.39	−0.01	0.03	0.71	0.86	2.35	−0.01
**NO**	field moist ^(1)^	Ntot [%]	0.03	0.74	0.85	2.21	−0.01	0.03	0.73	0.86	2.44	−0.01
** *YES* **	sieved_EPO_s-field_EPO_s	Ntot [%]	0.04	0.52	0.68	1.61	−0.01	0.05	0.33	0.69	1.55	−0.01
** *NO* **	lab_sieved-field moist	Ntot [%]	0.39	−35.28	0.03	0.19	−0.38	0.15	−5.57	0.06	0.5	−0.14
** *YES* **	fine_EPO_f-field_EPO_f	Ntot [%]	0.06	0.14	0.53	1.2	−0.02	0.04	0.62	0.81	2.06	−0.01
** *NO* **	lab_fine-field moist	Ntot [%]	0.77	−145.36	0.01	0.09	−0.77	0.15	−6.16	0.07	0.48	−0.15
**YES**	sieved_EPO_s	SOC [%]	0.23	0.78	0.88	2.13	−0.05	0.24	0.69	0.84	1.59	−0.02
**NO**	lab_sieved	SOC [%]	0.23	0.78	0.86	2.13	−0.03	0.25	0.68	0.86	1.54	0.08
**YES**	fine_EPO_f	SOC [%]	0.22	0.79	0.88	2.17	−0.09	0.21	0.78	0.87	1.86	−0.01
**NO**	lab_fine	SOC [%]	0.22	0.79	0.88	2.18	−0.08	0.22	0.76	0.87	1.8	0.07
**YES**	field_EPO_s	SOC [%]	0.29	0.66	0.82	1.71	−0.04	0.34	0.4	0.75	1.13	−0.05
**NO**	field moist ^(1)^	SOC [%]	0.22	0.8	0.89	2.23	−0.05	0.25	0.68	0.84	1.56	−0.09
**YES**	field_EPO_f	SOC [%]	0.21	0.82	0.89	2.33	−0.08	0.3	0.55	0.8	1.3	−0.04
**NO**	field moist ^(1)^	SOC [%]	0.22	0.8	0.89	2.23	−0.05	0.25	0.68	0.84	1.56	−0.09
** *YES* **	sieved_EPO_s-field_EPO_s	SOC [%]	0.33	0.53	0.7	1.45	−0.07	0.38	0.24	0.65	1.01	−0.10
** *NO* **	lab_sieved-field moist	SOC [%]	3.21	−42.07	0.03	0.15	−3.16	1.24	−6.93	0.05	0.31	−1.19
** *YES* **	fine_EPO_f-field_EPO_f	SOC [%]	0.45	0.16	0.56	1.09	−0.17	0.31	0.51	0.77	1.26	−0.12
** *NO* **	lab_fine-field moist	SOC [%]	6.41	−170.46	0.01	0.08	−6.35	0.65	−1.2	0.16	0.59	0.54

^(1)^ The field moist uncorrected datasets are the same, so the values are repeated.

## Data Availability

Raw data are available on request.

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
