# Peer review of "Prediction Accuracy of Soil Chemical Parameters by Field- and Laboratory-Obtained vis-NIR Spectra after External Parameter Orthogonalization"

_sensors, 2024, doi:10.3390/s24113556_

Round 1

Reviewer 1 Report

Comments and Suggestions for Authors

The article “Prediction accuracy of soil chemical parameters by field- and laboratory-obtained vis-NIR spectra after external parameter orthogonalization” is an interesting study in current modern topics on the use of mathematical methods to improve the analytical capabilities of long-known physical and chemical methods. The article is well written and worked out from the point of view of mathematical data analysis, but the study should be strengthened in terms of the analysis methods used. The weakest point of the article is obviously the lack of analysis of the obtained visible and IR spectra. It is obvious that the spectra of soil samples contain a similar set of absorption bands corresponding to certain chemical groups. By selecting characteristic absorption bands, the task of identifying soils could be significantly simplified. For example, the explanation of the effect of water on the content of clays in soils (pp. 398-401) is obviously related to the overlap of absorption bands of carboxyl groups, etc.

Other notes:

It is necessary to consider in more detail other methods for identifying adsorption spectra, and first of all, artificial neural network technologies. It is necessary to more clearly indicate the advantages of the approach to processing spectra used in this work compared to those described in other studies.

For better understanding of readers, it is advisable to provide the section on the analysis of soil samples with a graphical diagram of the research being carried out.

In Figure 3a, the optical spectra fall outside the standard definition limit of spectrophotometers (2 optical units). What is the error in determining optical density in this range?

Reviewer 2 Report

Comments and Suggestions for Authors

This study utilized External parameter orthogonalization (EPO) to correct the soil FTIR signals, and the method demonstrated strong practicality. It is recommended for publication after minor revisions. Here are some detailed suggestions:

1. The Introduction should provide an overview of other correction methods, especially those for handling high moisture content samples.

2. Almost all axes titles in the figures lack units.

3. The FTIR spectra need to be labeled with the corresponding chemical assignments for the peaks.

4. Why is there such a significant difference in the EPO-corrected spectra between the two different spectrometers?

5. Some content in Figure 4 is repetitive with Table 3. Would it be sufficient to keep Table 3 only? If Figure 4 is to be retained, it is recommended to perform a significance analysis.

Reviewer 3 Report

Comments and Suggestions for Authors

This manuscript resolved the water challenge in predicting soil parameters in situ by spectroscopy using EPO.  Here, the difference matrix was established different from other studies, and the use of EPO to resolve the water challenge was also innovative.  Results were also ideal.  The work is novel and valuable. 

Round 2

Reviewer 1 Report

Comments and Suggestions for Authors

The authors have corrected all comments